



# Failed cyclogenesis of a mesoscale convective system near Cape Verde: The role of the Saharan trade wind layer among other inhibiting factors observed during the CADDIWA field campaign

Guillaume Feger[1], Jean-Pierre Chaboureau[1], Thibaut Dauhut[1], Julien Delanoë[2], and Pierre Coutris[3]

[1]Laboratoire d'Aérologie (LAERO), Université de Toulouse, CNRS, UT3, IRD, Toulouse, France
[2]Laboratoire Atmosphères, Milieux, Observations Spatiales (LATMOS), Sorbonne Université and Université Paris Saclay, CNRS, Paris, France
[3]Laboratoire de Météorologie Physique (LAMP), CNRS, Clermont-Ferrand, France

**Correspondence:** Guillaume Feger (guillaume.feger@univ-tlse3.fr)

**Abstract.** The role of the Saharan air layer in Cape Verde cyclogenesis remains uncertain. Here, we investigate the inhibiting factors leading to the failed cyclogenesis of the mesoscale convective system (MCS) Pierre Henri (PH) observed during two flights of the CADDIWA campaign. We use CADDIWA data and a convection-permitting simulation run with the Meso-NH model. We show that the African easterly wave in which PH is embedded forms a marsupial pouch that keeps the SAL away from PH. On the contrary, a dusty, dry and warm air layer between 0.8 and 2 km altitude, called Saharan trade wind layer (STWL), penetrates into PH convective core, increasing the convective inhibition (CIN) area, and contributing up to 40% of the CIN area during the MCS mature phase. The cold pools produced by convection also increase the CIN area, and contribute up to 50% of this area after the MCS intense phase. Upper tropospheric (UT) dry air, with relative humidity below 15% between 7 and 11 km altitude, gradually penetrates into the 150 km circle around PH, reaching 18% of the area during its dissipation phase, preventing the MCS anvil expansion. The inhibiting roles of the STWL, cold pools, and UT dry air in leading the cyclogenesis to fail provide new insights into the complex dynamics of cyclogenesis in the Cape Verde region and challenge the existing model of the SAL.

## 1 Introduction

From June to October, African easterly waves (AEWs) are synoptic-scale disturbances formed over sub-Saharan Africa (Burpee, 1972). They travel westward into the Atlantic Ocean, where they are responsible for more than half of all tropical cyclones (Landsea, 1993). Tropical cyclogenesis from AEWs often results from a pre-existing mesoscale convective system (MCS), a feature particularly observed in the Cape Verde region when MCSs are embedded in the trough of AEWs (Arnault and Roux, 2011). It also depends on many other environmental conditions, including warm ocean waters, moist convective instability, weak vertical wind shear, a relatively moist mid-troposphere, which are linked, in the eastern tropical Atlantic, to the West African monsoon flow, the African Easterly Jet (AEJ), AEW activity and Saharan Air Layer (SAL) outbreaks. These complex dynamics make accurate forecasting of the tracks and intensities of North Atlantic tropical cyclones a persistent challenge for numerical weather prediction (Wang et al., 2018).





The SAL originates from the uplift of the Saharan mixing layer during the boreal spring and summer and is characterized by warm, stable air with low relative humidity and high dust content (Carlson and Prospero, 1972). It spans from about 1.5 km up to 7 km in altitude and is transported by the AEJ toward the Atlantic (Karyampudi and Carlson, 1988). The SAL impact on cyclogenesis depends on multiple factors, alternately supporting or hindering its development, making it a subject of ongoing debate (Shu and Wu, 2009). The stronger baroclinicity along the SAL edges reinforces the meridional circulation necessary for the AEW growth (Karyampudi and Carlson, 1988; Karyampudi and Pierce, 2002). Negative correlation between SAL outbreaks, evidenced by high dust loading, and Atlantic tropical cyclone activity was found using satellite data (Evan et al., 2006), reanalysis (Xian et al., 2020), and fully coupled general circulation model (Strong et al., 2018). Using satellite imagery, Dunion and Velden (2004) proposed three mechanisms for the suppression of Atlantic tropical storm activity by SAL: (i) the intrusion of dry air into the MCS promoting convectively driven downdrafts that contribute to stabilize the atmosphere, (ii) the intensification of vertical wind shear caused by the SAL's midlevel easterly jet and (iii) the trade wind inversion reinforced by the radiative heating of dust that stabilizes the environment. Shu and Wu (2009) further showed that cyclone weakening occurs when the SAL was detected into the southwest quadrant within 360 km of the tropical cyclone center. Comparing intensifying and weakening storms, Braun (2010) demonstrated that the SAL hindered convection in areas not crucial for cyclone development, while confining it to vorticity-rich regions, resulting in little evidence for a negative impact of the SAL on cyclogenesis. The dust content of SAL also impacts MCS dynamics, convective activity and cloud microphysics (Luo and Han, 2021; Sun and Zhao, 2020; Pan et al., 2018). To analyze the impact of the SAL on deep convection over the Atlantic, air parcel theory provides a useful theoretical framework (e.g. Wong and Dessler, 2005). They showed that the warmness and dryness of the SAL raise the lifting condensation level (LCL) and level of free convection (LFC), thus reinforcing the energy barrier for moist convection to occur.

A second synoptic feature inhibiting convection during cyclogenesis is the intrusion of dry air into the free troposphere. When considering the intrusion of dry air over the Eastern Atlantic, this is often the intrusion of the low-to-mid level SAL into the cyclone (e.g., Dunion and Velden, 2004; Braun, 2010). In this case, the role of the marsupial pouch, a protected region of cyclonic recirculation that exists in the lower troposphere, is essential (Dunkerton et al., 2009). For example, Fritz and Wang (2013) examined the impacts of dry air on the formation of two tropical cyclones. They found the suppression of deep convection by mid-level drying for the non-developing cyclone where a well-defined wave pouch is absent, and dry air at the pouch periphery remaining away from the pouch center for the developing cyclone. The intrusion of dry air in the upper troposphere (UT) did not attract as much attention as midlevel dry air intrusion. Hankes et al. (2015) found the presence of mid- and upper-level dry air in non-developing waves in the Cape Verde region. They attributed the dry air to a broad trough leading to strong westerly vertical shear, which is detrimental to tropical cyclone formation. Arnault and Roux (2011) also found inhibition of non-developing waves by unusually dry environment and strong vertical wind shear in the same region.

Cold pools produced by deep convection also play an important role in storm development. Cold pools are regions of relatively cold air at the surface, spanning from 10 to 200 km in diameter (Zuidema et al., 2017), originating from the evaporation of rain from convective downdrafts (Charba, 1974). Their expansion can lift warmer ambient air aloft, potentially generating new convective cells (Goff, 1976; Tompkins, 2001; Schlemmer and Hohenegger, 2014). Cold pools modify near-surface ther-





modynamics (Zuidema et al., 2017). Inside cold pools, the cold air brought in by downdrafts stabilizes the atmosphere causing a local suppression of convection. The duration of this convective inhibition depends on the time taken for surface fluxes to remove the negative temperature perturbation (Tompkins, 2001).

The aim of the present study is to assess the role of SAL, cold pools and UT dry air on tropical cyclogenesis. To achieve this goal, we take advantage of the Clouds-Atmospheric Dynamics-Dust Interactions in West Africa (CADDIWA) field campaign that took place in September 2021 (Flamant et al., 2024). The CADDIWA project objectives were to investigate dust effects on AEWs dynamics and tropical storm formation over the North Atlantic. Here, we focus on the case of a non-cyclogenesis of a MCS named Pierre Henri (PH) sampled by two flights during the campaign. Jonville et al. (2024) analyzed three AEWs, including the one in which PH was embedded. Despite the presence of a marsupial pouch developed in the AEW trough, they suggest that the SAL intrusion inside the pouch led to inhibition of PH cyclogenesis. Our study builds upon this synoptic analysis by examining the SAL intrusion in the convective core of PH at mesoscale. By incorporating observational data from CADDIWA and considering secondary inhibiting factors such as cold pools and UT dry air, it provides a detailed assessment of why MCS PH did not develop into a tropical cyclone. We take advantage of a convection-permitting simulation made with the Meso-NH model to analyze the aerosols and thermodynamics of the SAL, as well as the transport of Saharan air into the MCS. In particular, we describe the Saharan trade wind layer (STWL) beneath the SAL and show its major role in inhibiting the formation of MCS PH into a tropical cyclone.

The study is structured as follows: Sect. 2 describes the data. Sect. 3 details the synoptic context, the MCS lifecycle and the atmospheric characteristics upstream of the MCS, including STWL, SAL and UT dry air. Sect. 4 analyzes the contribution of the inhibition factors, STWL, cold pools and UT dry air, to MCS PH dissipation. Sect. 5 gives the conclusions.

## 2 Data and methods

### 2.1 Meso-NH simulation

The simulation is run with the Meso-NH nonhydrostatic model version 5.6 (Lac et al., 2018), on a domain of size 1730 km × 1350 km, encompassing the West African coast and Cape Verde Islands (Fig. 1). The horizontal grid spacing is set to 3 km allowing the model to resolve deep convection explicitly. The vertical grid has 92 levels with a grid spacing stretched from 30 m near the surface to 500 m from 9 km up to the top of the model at 26 km. The simulation is run for 24 h, starting at 00:00 UTC on 11 September 2021, with output saved every hour. Initial and 6-hourly boundary conditions are obtained from the European Centre for Medium-Range Weather Forecasts (ECMWF) operational analysis for surface and meteorological variables and from the Copernicus Atmospheric Monitoring Service (CAMS) reanalysis (Inness et al., 2019) for aerosol distribution. The CAMS reanalysis includes aerosol mass mixing ratios over 60 vertical levels and assimilates aerosol optical depth (AOD) retrievals from Moderate Resolution Imaging Spectroradiometer (MODIS).

The simulation uses the Surface Externalisée (SURFEX) scheme for surface fluxes (Masson et al., 2013), the radiative scheme taken from ECMWF (Gregory et al., 2000), and an eddy-diffusivity mass-flux scheme for shallow convection (Pergaud et al., 2009). The turbulence is parameterized using the 3D mode of a 1.5-order closure scheme (Cuxart et al., 2000), to





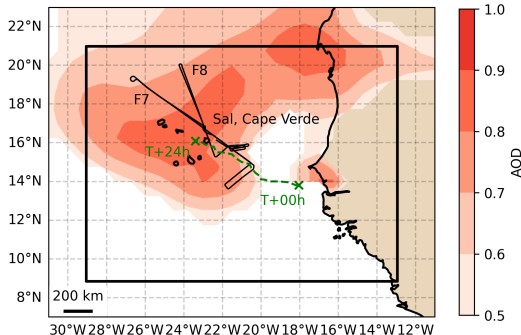

**Figure 1.** Simulation domain (black rectangle), which includes the Cape Verde Islands and a portion of the West African continent (shaded brown). The morning (F7) and afternoon (F8) flight tracks are depicted with black lines. The MCS trajectory is represented by the green dashed line. Red shading shows the aerosol optical depth greater than 0.5, as derived from the CAMS reanalysis at 00:00 UTC on 11 September 2021.

obtain a better representation of cloud organization and lifetime (Machado and Chaboureau, 2015). The cloud microphysics is the two-moment LIMA (Liquid Ice Multiple Aerosols) scheme (Vié et al., 2016) which predicts both the concentration and mixing ratio of five hydrometeor species (cloud droplets, raindrops, ice crystals, snow and graupel particles). LIMA is driven by the prognostic evolution of a three dimensional aerosol population composed of cloud condensation nuclei (CCN) and ice 95 freezing nuclei (IFN). Aerosols nucleating properties are used for the formation of cloud droplets and ice crystals. A simplified initialization of the aerosol population is used, consisting of one type of CCN and one type of IFN. The CCN concentration is initialized from the sum of CAMS sea salt mode contents, while the IFN concentration is derived from the sum of dust mode contents. Aerosol depletion by the activation process is disabled in LIMA to avoid any limiting impact on convection. To assess convective activity and cloud organization in the simulation, synthetic brightness temperatures (BTs) at $10.8\,\mu$m are generated 100 from model outputs using the radiative transfer model for the TIROS Operational Vertical Sounder (RTTOV) code (Saunders et al., 2018), which has been integrated into Meso-NH by Chaboureau et al. (2008).

## 2.2 Observations

The French Falcon 20 aircraft, operated by SAFIRE (Service des Avions Français Instrumentés pour la Recherche en Environnement) for environmental research, was extensively used during the CADDIWA campaign. Detailed information about 105 the Falcon 20 and its research payload during the CADDIWA mission is available in Flamant et al. (2024). This study uses data from two of the nine research flights of the campaign, flights F7 and F8, conducted in the morning and afternoon on 11 September 2021. In the following, flight tracks and instrument payload are detailed.

Flight F7 aimed to study the dry air and dust circling ahead of MCS PH, as well as its anvil. The Falcon 20 departed Sal at 09:15 UTC, flew toward the SAL at 4 km altitude, and turned back at 10:10 UTC toward PH at 10 km altitude and landed 110 at 12:43 UTC. Vaisala dropsondes were released, three before reaching the MCS (D1, D2, D3) and two more in the MCS (D4



and D5). Flight F8 targeted the same system adding transverse legs at different altitudes to sample the convective system. This flight had a more north-south orientation compared to F7 to capture the core of the convection. The Falcon 20 took off at 14:46 UTC, turned back towards the SAL at 15:38 UTC, and landed at 17:56 UTC. Eight dropsondes were released, including one in the UT dry air north of the MCS (D6).

The Falcon F20 thermodynamic probes include the Rosemount T for temperature and the FLYHT WVSS-2 for humidity. The microphysical probes set is composed of the ultra-high-sensitivity aerosol spectrometer (UHSAS-A), the forward-scattering spectrometer probe (SSP-300), the cloud droplet probe (CDP-2) and 2D-S Stereo Probe. The UHSAS, the SSP-300 and the CDP probes are optical-scattering aerosol particle spectrometers that counts and sizes aerosol particles and clouds droplets. The size of each particle is derived from the scattered intensity using Rayleigh (40-300 nm) or Mie (0.3-50 $\mu$m) scattering models.

The ranges of the 3 probes are 0.04-1 $\mu$m, 0.3-20 $\mu$m, 2-50 $\mu$m, respectively. The 2D-S is an optical array probe imager in the 10–1280 $\mu$m size range implying 128 photodiodes recording silhouette of particles at high frequency (up to 20 MHz).

The Falcon F20 carries a combination of lidar and radar (RALI, Delanoë et al., 2013) instruments to remotely sense the vertical structure of aerosols and clouds . The RALI platform combines the radar RASTA (RAdar SysTem Airborne) and the LNG lidar (Léandre new-generation). The RASTA is a multibeam 95 GHz Doppler cloud radar with a range resolution of

60 m. For the CADDIWA campaign, it was composed of three downward-looking antennas and one upward-looking antenna with an integration time of 250 ms each, leading to a measurement on the same antenna every second at 200 m s$^{-1}$. Thanks to the Doppler capability and the three downward-looking antennas, the three-dimensional wind field below the plane can be retrieved. The cloud mask is generated by filtering out background noise from the radar signal for each antenna using a thresholding technique and image processing to remove isolated pixels. LNG is a triple-wavelength (355, 532 and 1064 nm)

dual-polarization lidar. It includes depolarization at 355 nm and has a vertical resolution of 37 m. Considering the calibrated signal is averaged over 5 s, the horizontal resolution is 1 km for an aircraft flying at 200 m s$^{-1}$, on average. Here we analyze the backscatter coefficient at 532 nm and take advantage of the RALI phase categorization product to detect clear sky.

This study also incorporates space-borne observations, utilizing infrared imagery at a wavelength of 10.8 $\mu$m obtained from the Spinning Enhanced Visible and InfraRed Imager (SEVIRI) aboard the Meteosat Second Generation geostationary satellite

to analyze cloud population evolution, and the MODIS DeepBlue aerosol optical depth (AOD) product at 500 nm to evaluate the horizontal extent of the dust outbreak.

## 3 Overview of the MCS PH lifecycle and its environment

### 3.1 Synoptic context

MCS PH evolved in a synoptic environment characterized by Saharan air to the north, a monsoon flow to the south, and between

the two, the AEJ whose position is modulated by an AEW. The AEW was first identified over the West African continent on 8 September. It propagated westward, reaching the coast south of Dakar on 10 September, weakened before crossing the coast, and redeveloped over the Atlantic Ocean at 00:00 UTC on 11 September. This redevelopment led to the formation of MCS PH within the trough of the AEW, at the southern edge of the AEJ. The NOAA National Hurricane Center (NHC) had



initially forecasted that this intense MCS would develop into a tropical storm, but downgraded it to a tropical disturbance on

11 September so it is not recorded by the NHC (Flamant et al., 2024). MCS PH took a westward trajectory until 12:00 UTC on 11 September, after which it unexpectedly moves northwest direction. Throughout the day, the monsoon penetration behind the through is intense, supplying moisture and vorticity to the low levels of the MCS. The AEJ intensity progressively weakens until the MCS dissipates around 00:00 UTC on 12 September near Sal Island in Cape Verde. The SAL also played a significant role in shaping the atmospheric conditions near Cape Verde. The SAL reached the Cape Verde Islands on 9 September, raising

AOD 500 nm values from 0.18 to 0.47, and further to 0.75 on 10 September at the Aerosol Robotic Network (AERONET) Cape Verde station at Sal. On 10 September, MODIS AOD exceeded 0.9 between 14° N and 22° N at the Cape Verde longitudes. On 11 September, the dust plume covered over half of the simulation domain, as shown by CAMS AOD (Fig. 1). On 12 September, the area with AOD values above 0.9 shifted northwest of Cape Verde.

## 3.2    Lifecycle of MCS PH

The evolution of PH on 11 September and its environment is briefly described. The occurrence of deep convective clouds from observations and simulation is analyzed at four keys times on 11 September (Fig. 2). Brightness temperatures at 10.8 $\mu$m between 210-230 K (in dark green) are classified as deep convective clouds (DCCs), and those below 210 K (in light green) as high convective clouds (HCCs). Dust content is assessed using MODIS AOD measurements at 11:50 UTC and simulated mean dust concentrations between the surface and the top of the dust plume at a height of 6 km at 11:00 UTC (Fig. 2c, d).

At 06:00 UTC, the spatial distribution of HCCs and DCCs shows that the MCS is compact and circular in shape in both observations and simulation. The extent of HCCs area exceeds 100 km$^2$, marking an intense convective activity (Fig. 2a, b). At 11:00 UTC, it reduces to about 20 km$^2$, showing a strong decrease in convective activity (Fig. 2c, d). Scattered convective cells are observed and simulated to the south-east of the MCS. In the model, HCCs shift northward, close to the leading edge of the MCS. The MCS is surrounded to the north-west by the SAL that contains a large amount of aerosols, as shown by MODIS

AOD values up to 1 and mean dust concentration values above 25 cm$^{-3}$ between 0 and 6 km height. At 16:00 UTC, the MCS reaches its maximal extent of DCCs (Fig. 2e, f). HCC activity is mainly observed and simulated close to the MCS leading edge, and reduces in the southern part of the MCS. The DCC extent increases and a discontinuity between the northern active zone and the southern sector of the MCS appears. At 21:00 UTC, the MCS passes over the Cape Verde Islands (Fig. 2g, h). HCCs are confined to a small area, and DCCs dissipate in both observations and simulation. Although the model accurately captures

this decay, the simulated MCS is located approximately 2 degrees south of its observed position. Warmer cloud formations are observed as spiral bands extending southward from HCCs, covering a larger area in the observations than in the simulation. Overall, the MCS shows a northwesterly track and decreasing activity at the end of 11 September, two features that are globally well reproduced by the simulation.

## 3.3    Atmospheric properties ahead of the MCS PH

In this section, we use data from F7 to describe the atmosphere north-west of the MCS. The main result is the presence of hot, dry and dusty Saharan air at the trade wind levels in the vicinity of the MCS. This layer will be referred to as the Saharan trade



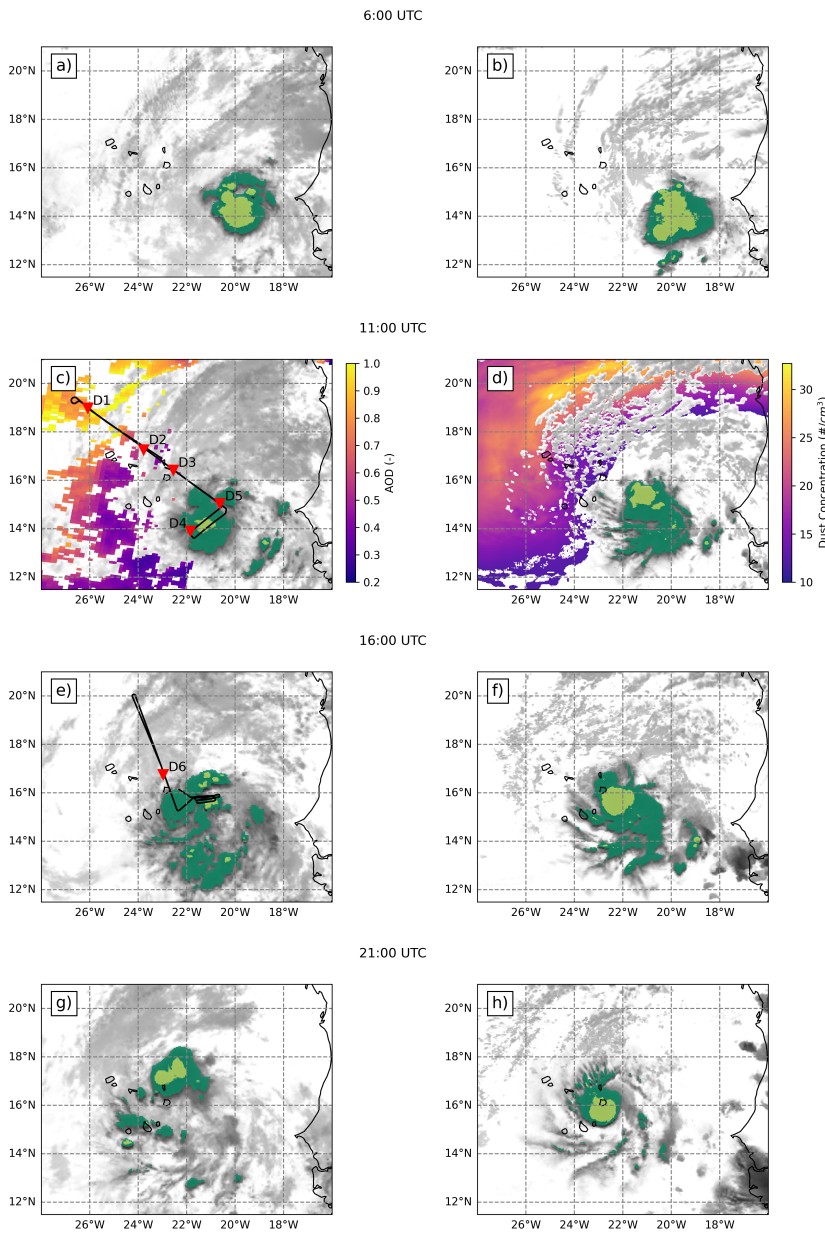

**Figure 2.** Brightness temperature at $10.8\,\mu$m from Meteosat-11 (left column) and the simulation (right column) at **(a, b)** 09:00, **(c, d)** 11:00, **(e, f)** 16:00 and **(g, h)** 21:00 on 11 September 2021. Deep convective clouds are shown in dark green (temperatures between 210 and 230 K) and light green (temperatures below 210 K). In **(c, d)**, colored shading shows AOD at 550 nm from MODIS Deep Blue at 11:50 UTC and mean dust concentration between 0 and 6 km altitude at 11:00 UTC, respectively. In **(c, e)**, the black line shows the morning (F7) and afternoon (F8) flight tracks, respectively. In **(c)**, the red triangles show the location of dropsondes D1 to D6.





wind layer (STWL). First, we describe the thermodynamics of the atmosphere between the SAL and the MCS. Second, we analyze the origins of SAL and STWL along the northwestern part of the F7 track. Third, we assess the vertical structure of the spatial distribution of aerosols. Finally, we examine the aerosol size distributions, from sea level to the top of SAL, to assess
their content in dust.

The vertical structure of the troposphere sampled by dropsondes D1, D2, and D3 shows temperature and humidity inversions from the SAL to the nearby MCS environment (Fig. 3). The three profiles share common features in the marine boundary layer (MBL) and the trade wind layer (TWL). They differ above 2 km height where D1 and D2 exhibit characteristics of the SAL, while D3 indicates no SAL at these altitudes but provides insight into the wind circulation around the MCS. The first layer
above sea level is a well-mixed MBL with a potential temperature of 299 K (Fig. 3a, f, k) and a vapor mixing ratio of 16–17 g kg$^{-1}$ (Fig. 3b, g, l). The top of this layer is identified by the largest humidity drop (von Engeln et al., 2005) and coincides with the first inversion layer. The MBL wind direction is N-NE, except for D3, where it may be affected by the topography of Sal Island (Fig. 3o). The second layer is the STWL, from 0.8 to about 2 km height. It lies above the MBL and its top is either bounded by the SAL (on D1 and D2) or, in the absence of SAL, by a temperature inversion (on D3 and during takeoff, as probed
by onboard aircraft measurements, not shown). The STWL is a stable stratified layer with large potential temperature gradients (6–12 K km$^{-1}$) (Fig. 3a, f, k) and dry air (3–12 g kg$^{-1}$) (Fig. 3b, g, l). Its wind direction ranges from E (Fig. 3e) to S-E (Fig. 3j, o). The closer it is to the MCS, the moister and the cooler it is. The third layer, observed on D1 and D2 only, is the SAL. It is a well-mixed neutral layer extending up to 4.8 km height, with a constant potential temperature of 317 K, a vapor mixing ratio of 4–6 g kg$^{-1}$ and E-NE winds ranging from 12 to 18 m s$^{-1}$ (Fig. 3d, e, i, j). On D3, 200 km from the MCS, an intense circulation
is evidenced by wind speed of 20 m s$^{-1}$ and relative humidity over 90 % around 4 km height. In D1 and D2, the fourth layer, between 4.8 and 7 km, shows a potential temperature increase from 317 to 325–333 K (Fig. 3a, f). The wind speed ranges from 12 to 18 m s$^{-1}$ and exhibits greater variability, with a direction shift eastward (Fig. 3d, e, i, j). The fifth layer, above 7 km height, appears on D1 and D2. It is characterized by vapor mixing ratio close to zero (Fig. 3b, g), a sharp positive gradient in potential temperature (Fig. 3a, f), low wind speeds and low vertical shear (Fig. 3d, i). The characteristics of this UT dry air indicates the
presence of a stable layer and the absence of an upper-level trough. The model reproduces the thermodynamics observed by D1, D2, and D3 but smooths the profiles, likely due to its coarse vertical resolution. Stronger wind speeds are simulated at D3 location due to the closer proximity of the simulated MCS at this location, whereas the observed MCS is further away (Fig. 2c, d).

The origins of the STWL and SAL are identified from 4.5 days back-trajectories calculated along the northwestern track
of F7 at heights of 1.5 and 3.2 km (Fig. 4) using Meso-NH forecasts made during the CADDIWA campaign (Flamant et al., 2024). Close to Sal Island, in the southern part of the dust outbreak, the STWL originates from southern Sahara, between 1 and 3 km, and gradually descend below 2 km before crossing the coast. The westernmost STWL air parcels arriving along the F7 track originate from an anticyclonic circulation at 25° N, and gradually descend from 5 to 1.5 km (Fig. 4a). The SAL originates from northern Algeria and travels across the continent toward the Atlantic in the AEJ at altitudes between 2 and 4 km (Fig. 4b).
Back-trajectories along the F7 track show that the STWL is mostly composed of Saharan air, but it differs from the SAL in that it mixes air from a wider variety of sources, resulting in higher stratification and greater variability in thermodynamic







**Figure 3.** Profiles of **(a, f, k)** potential temperature, **(b, g, l)** water vapor mixing ratio, **(c, h, m)** relative humidity, **(d, i, n)** wind speed and **(e, j, o)** wind direction from the dropsondes (black) launched at 10:20 (top, D1), 10:34 (middle, D2) and 11:04 UTC (bottom, D3) and from the simulation (orange) at 10:00 (top and middle) and 11:00 UTC (bottom) 11 September 2021. The blue horizontal line marks the top of MBL.



properties. Visual inspection of the CAMS reanalysis of the AOD the week before F7 confirms the convergence of two distinct dust plumes near the coast the day before. Consistent with the Meso-NH backtrajectories, one of the plumes originates from northern Algeria, while the other originates from the Bodélé region and propagates westward across the southern Sahara. Both

regions are well documented in the literature as major dust source areas (Karyampudi et al., 1999).

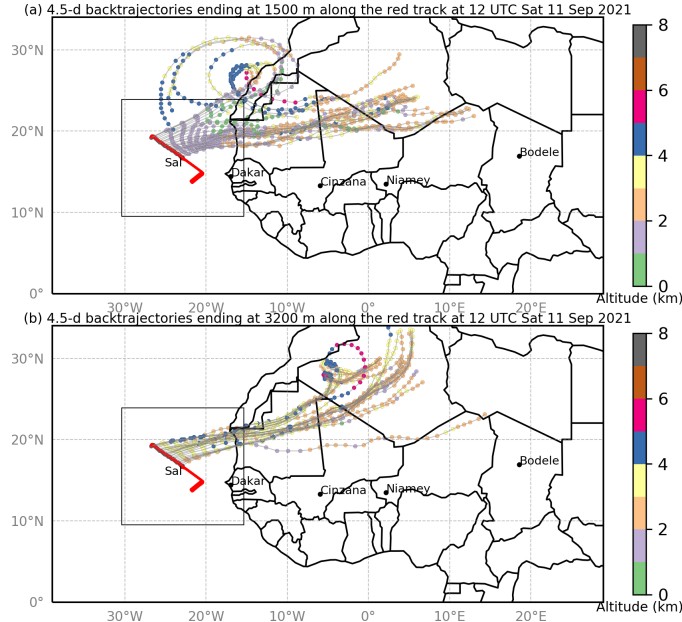

**Figure 4.** Back-trajectories of 4.5 days ending at 12:00 UTC 11 September 2021 at **(a)** 1500 and **(b)** 3200 m along the F7 track. Dots represent the evolution of the back-trajectories every 3 h.

The vertical profile of the troposphere along the northern part of the F7 track is further assessed with aerosol data from LNG, along with cloud and wind speed measurements from RASTA and dropsondes (Fig. 5a). Clouds in the MCS are indicated by RASTA reflectivities larger than -15 dBZ extending up to a height of 14.3 km. Horizontal wind speed measurements from D3 and Doppler radar show values ranging from 0 to $4 \, \mathrm{m \, s^{-1}}$ below 2 km height in the MCS and its surrounding environment.

Within the first 90 km, two plumes of aerosols in the STWL are observed with clear air above. From 90 to 860 km, LNG backscatter coefficients at 532 nm reveal a vertical stratification of the aerosol loading that matches with the MBL, STWL and SAL seen in D1 and D2. At 1050 km, logarithm of backscatter coefficient is below -5.5 $\log(\mathrm{m^{-1} \, sr^{-1}})$, suggesting the absence of aerosols at SAL levels in consistency with the absence of neutral layer observed in D3. From 2010 km to the end of the flight, aerosols are detected below 2 km height, in the vicinity of Sal Island. Simulated dust concentrations along the

F7 track are compared to observed backscatter coefficients (Fig. 5b). From 0 to 90 km, the STWL dust concentration above $5 \, \mathrm{cm^{-3}}$ extends up to 2 km height, with clear air above. The SAL dust plume appears at 90 km and vanishes after 860 km which is consistent with observations. The dust concentration in the MBL is always greater than $5 \, \mathrm{cm^{-3}}$. The simulated STWL dust distribution shows a larger concentration of aerosols at an altitude of 1 km compared to observational data. Simulated



and sampled aerosols show that the SAL is 300 km from the MCS, while the STWL is close to the MCS due to low wind
speeds below 2 km height. The simulated wind circulation is consistent with dropsonde and radar observations. Finally, the
simulation shows the ubiquitous presence of aerosols below 2 km in altitude along the F7 track, notably in areas that have
not been accessed or probed by CADDIWA aerosol instruments. It thus reveals the presence of aerosols beneath PH. The
consistency between observations and the simulation obtained along the F7 track gives us confidence in the realistic presence
of dusty layers in the first 2 km inside PH.

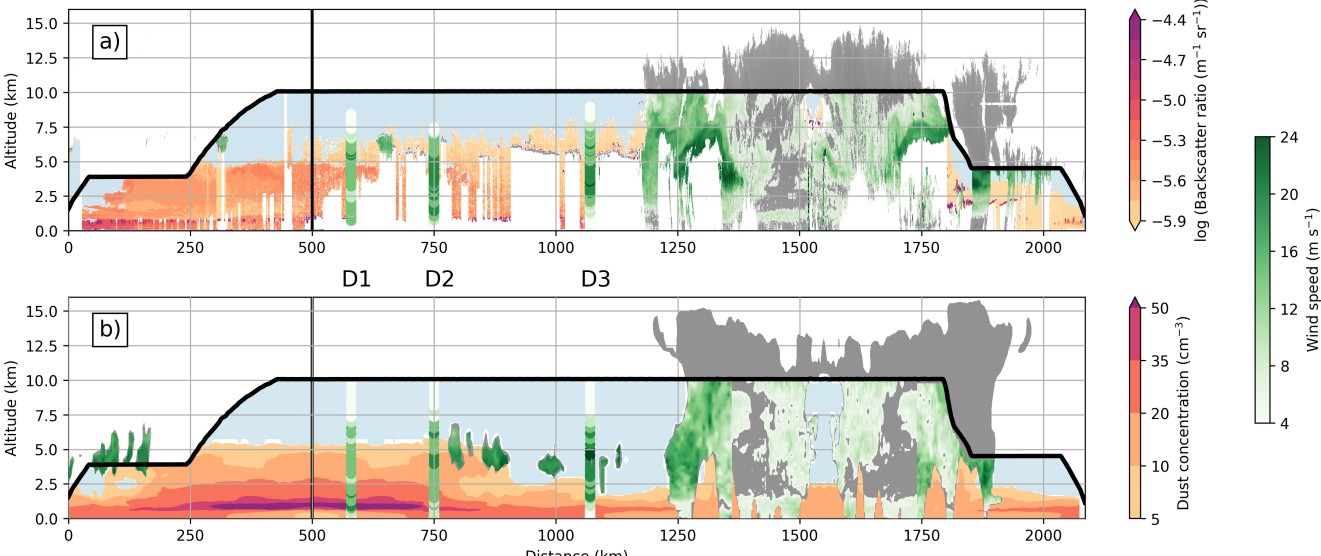

**Figure 5. (a)** Vertical cross-section of the logarithm of backscatter coefficient at 532 nm from LNG (orange shading), reflectivity beyond
-15 dBZ and supercooled water (gray), and Doppler wind speed above $4\,\mathrm{m\,s^{-1}}$ (green shading) from RASTA along the F7 track. Wind speeds
from dropsondes D1, D2, and D3 appear in green. Clean sky from RALI observation appears in blue. **(b)** Vertical cross-section of simulated
dust concentrations beyond $5\,\mathrm{cm^{-3}}$ (orange shading), reflectivity beyond -15 dBZ (gray), wind speeds above $4\,\mathrm{m\,s^{-1}}$ in cloud and at D1, D2,
D3 and D4 locations along the F7 track. The F7 track appears in thick black lines and the u-turn as a vertical line at 500 km. Observations
were taken between 09:17 to 11:40 UTC and simulation outputs at 10:00, 11:00 and 12:00 UTC.

Additional measurements of aerosol plumes are obtained from optical particle counters onboard the aircraft. Aerosol size
distributions are averaged in five regions (Fig. 6): the MBL (between 0 and 800 m height at 09:15 UTC), the STWL between
1 and 2 km height during take-off (09:17 UTC) and landing (12:38 UTC), the SAL (at 4 km height from 09:17 to 09:35 UTC),
and a clean air layer (between 2 and 3.2 km height at 09:18 UTC). On the one hand, STWL aerosol concentrations at take-off
and landing have the same size distribution. These measurements, taken more than three hours apart, suggest the persistent
properties of aerosols in the STWL. On the other hand, the size distribution of aerosols above $0.3\,\mu$m is similar in the MBL,
STWL and SAL compared with the clean air layer. In particular, these layers show a similar size distribution for the two
accumulation modes, around 0.4 and $0.9\,\mu$m. Assuming that the SAL contains mainly dust aerosols consistent with the back-



trajectories shown in Fig. 4, this suggests the presence of dust in both the STWL and MBL. These three dust layers differ, however, in their size distribution for the fine and coarse aerosol modes. First, fine aerosol concentrations are relatively similar
in SAL and the clean layer, whereas they are three times higher in the MBL and STWL. Second, coarse aerosol concentrations ($>1\,\mu$m) are up to 10 times higher in the MBL than in the SAL and STWL, and up to 100 times higher than in the clean air layer (which is probably due to the presence of sea salt). Previous studies have documented the presence of dust within the MBL (Colarco et al., 2003), and the complexity of routes by which dusty air reaches a tropical cyclone near Cape Verde (Schwendike et al., 2016) but none provide the multi-instrumental observational evidence of multiple layers of dust in the vicinity of a MCS
shown here.

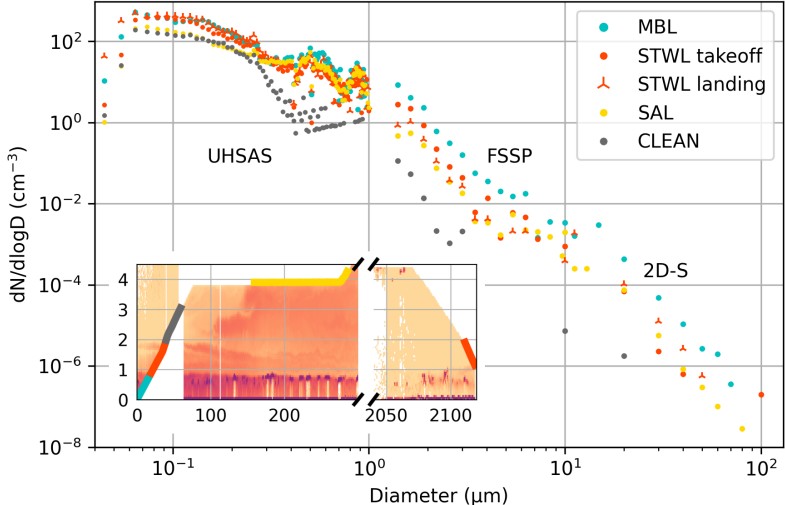

**Figure 6.** Mean size distributions measured by the UHSAS, FSSP and 2D-S microphysical probes over the five regions defined in the inset: MBL (blue), STWL (red), clear air (gray), and SAL (gold). In the insert, the red shading is the same as used in Fig. 5a.

The vertical distribution of aerosols in the MCS environment can be more accurately assessed through the synthesis of satellite data and CADDIWA observations. Within 100 km west of the MCS, CADDIWA measurements show that aerosol plumes are confined to altitudes below 2 km along the F7 track, and MODIS AOD values show moderate aerosol loading with values between 0.3 and 0.6 extending from 12 to 18° N (Fig. 2c). Even if the SAL does not interact directly with the MCS,
it may be associated with a STWL that can affect the MCS. This highlights the need to characterize SAL edge properties, in particular the vertical distribution of aerosols. It also challenges the realism of the conceptual model of the summer SAL, often described as a dry, well mixed layer above a relatively cool and moist trade wind layer that confines the SAL to altitudes above 1.2-1.8 km (Prospero and Carlson, 1972). Lastly, this analysis can provide insights to understand the unusual configuration observed during a major dust event where large dust loading are observed above the MBL and below 2 km height (Marenco
et al., 2018).





## 4  STWL, cold pools and UT dry air impacts on MCS PH lifecycle

The impacts of convective inhibition factors on MCS PH lifecycle are analyzed in the vicinity of its convective core. The method applied here consists in (i) defining a critical area, (ii) identifying inhibition factors in this area and (iii) analyzing their variation, and that of the MCS activity, over time. The critical area is defined following the framework of Shu and Wu (2009),

which identified a critical radius of 360 km centered on tropical cyclones within which SAL air intrusion inhibits cyclogenesis. As MCS PH is smaller than tropical cyclones, we employed a radius of 150 km, centered on the convective core associated with HCCs and referred to as D150. In the following, we show that STWL, cold pools and UT dry air inhibit PH development. Of these three convection inhibition factors, only STWL and cold pools have an impact supported by air parcel theory. For that purpose, convective available potential energy (CAPE) and convective inhibition (CIN) are computed as vertically integrated

measures of the buoyant energy for an air parcel ascending from 50 m above the surface. In particular, a CAPE of $1000 \, \mathrm{J \, kg^{-1}}$ indicates a tropical atmosphere capable of sustaining convective activity (Rennó and Ingersoll, 1996), while a CIN of $1 \, \mathrm{J \, kg^{-1}}$ is a barrier that can be overcome by an air parcel with a vertical wind speed of at least $1.4 \, \mathrm{m \, s^{-1}}$ below the level of free convection (Reinares Martinez and Chaboureau, 2018).

### 4.1  STWL, cold pools and UT dry air at four key times

Simulated horizontal wind circulation and dust concentrations above $15 \, \mathrm{cm^{-3}}$ are analyzed in the vicinity of the convective cells of the MCS using horizontal cross sections at the MBL, STWL and SAL levels at 11:00 UTC (Fig. 7). HCCs and DCCs associated to the MCS are also shown (Fig. 7c). At 500 m height, streamlines originating from the west and south converge within the northern part of the MCS, transporting dust-laden MBL air from the northwest and west (Fig. 7a). At 1.5 and 4 km heights, curved streamlines mark the AEW in which the MCS is embedded. At 1.5 km height, streamlines with speeds below

$10 \, \mathrm{m \, s^{-1}}$ and dust concentrations above $15 \, \mathrm{cm^{-3}}$ are present from the west up to the southwestern convective cells (Fig. 7b). An almost closed circulation pattern with an approximate diameter of 40 km is at the center of the convective cells, while a weak converging circulation to the southwest of the MCS facilitates the intrusion of the STWL into the nearby convective cells. Maximal wind intensity values, close to $15 \, \mathrm{m \, s^{-1}}$, reached northeast of the MCS. At 4 km height, streamlines originate from the south and the east (Fig. 7c). The wind intensity reaches its peak north of the MCS, with values surpassing $20 \, \mathrm{m \, s^{-1}}$

and remains steady westward. Dust concentration remains below $15 \, \mathrm{cm^{-3}}$ suggesting no interaction between the SAL and the MCS at 4 km height.

STWL intrusion into the MCS is investigated further with additional time and variable analyzes. Here, we identify properties of three convective inhibition factors: STWL, cold pools and UT dry air. STWL is identified by relative humidity below 80 % and dust concentration above $10 \, \mathrm{cm^{-3}}$, between 0.8 and 2 km height. (The 80% threshold corresponds to the value measured

by dropsonde D4 (Fig. 9a).) Cold pools are identified using a buoyancy threshold value, following Tompkins (2001) among others. They are defined as areas with buoyancy below the $-0.03 \, \mathrm{m \, s^{-2}}$ threshold, buoyancy being calculated as the fluctuation of the density potential temperature relative to the horizontal average at 50 m height in a $400 \, \mathrm{km^2}$ square visually centered on the MCS. The choice of the buoyancy threshold does not affect the main conclusion of the study. UT dry air is identified





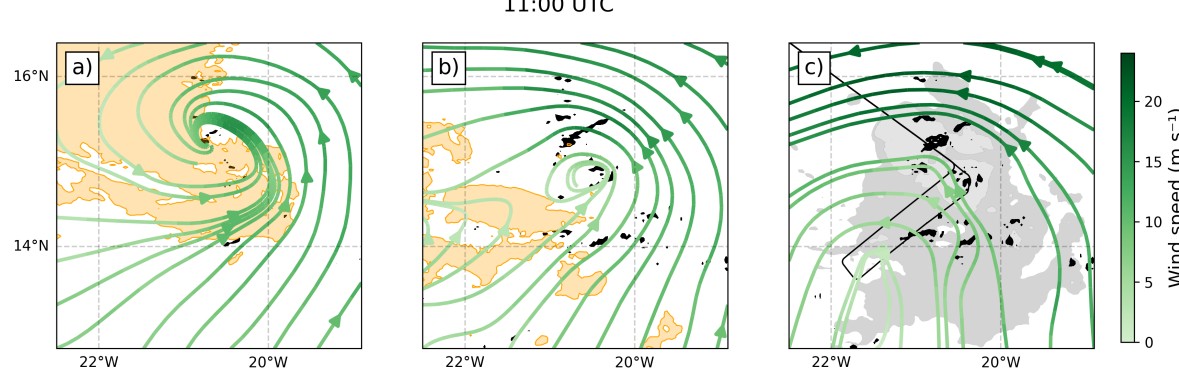

**Figure 7.** Horizontal cross sections at 11:00 UTC in the simulation. Horizontal wind speeds (green streamlines), dust concentrations (beyond $15\,\mathrm{cm}^{-3}$ in orange shading) and vertical wind speeds (larger than $1\,\mathrm{m\,s}^{-1}$ in black shading) at **(a)** 500, **(b)** 1500 and **(c)** 4000 m. In **(c)**, dust concentration remains lower than $15\,\mathrm{cm}^{-3}$, and brightness temperature between 210 and 230 K and below 210 K are shown in gray, and light gray respectively. The black line shows the F7 track of the F20 aircraft.

with mean relative humidity between 8 and 11 km height below 40%. Overlaid horizontal cross sections from the simulation at
06:00, 11:00, 16:00 and 21:00 UTC over a $400\,\mathrm{km}^2$ region visually centered on the MCS are shown (Fig. 8). The STWL and cold pools are shown along with D150 and low level circulation (Fig. 8a, d, g, j). Convective activity favoring and inhibiting factors are assessed using CAPE and CIN (Fig. 8b, e, h, k). The relative location of UT dry air, vertical wind speeds above $1\,\mathrm{m\,s}^{-1}$ at 9 km height, HCCs and DCCs are analyzed (Fig. 8c, f, i, l).

At 06:00 UTC, the STWL is to the west of the MCS, while cold pools are situated in the eastern part of D150 (Fig. 8a).
Streamlines show linear shaped convergence at low levels. CIN values above $1\,\mathrm{J\,kg}^{-1}$ appear in the northwestern quarter and above the cold pools, while CAPE is above $1500\,\mathrm{J\,kg}^{-1}$ in the eastern portion of the domain, except above the cold pools and west of the southern convective cells (Fig. 8b). Convective cells extend up to 9 km height and moisten the UT dry air layer to the south west (Fig. 8c). At 11:00 UTC, patches of STWL from the southwest reaches the D150 area (Fig. 8d). The low level circulation becomes cyclonic. The area covered by cold pools increases to the southwest of the MCS. It is partly collocated
with CIN values above $1\,\mathrm{J\,kg}^{-1}$ (Fig. 8e). A stripe of CIN above $1\,\mathrm{J\,kg}^{-1}$ extends into D150 from the west, coinciding with CAPE values below $1000\,\mathrm{J\,kg}^{-1}$. CAPE above $2000\,\mathrm{J\,kg}^{-1}$ appears to the northeast and southeast of the MCS, while UT dry air appears northwest of the MCS (Fig. 8f). Convective cells at 4 km height are more scattered, while at 9 km height they weaken, except in the northern region of the MCS (Fig. 8e,f). At 16:00 UTC, convective cells aggregate to form a core where low-level air converges (Fig. 8g). A few convective cells appears to the south of the MCS, outside of D150. The STWL continues to
intrude to the south and west of D150. The cold pools extension is reduced in D150, while the CIN zone above $1\,\mathrm{J\,kg}^{-1}$ and the CAPE zone below $1000\,\mathrm{J\,kg}^{-1}$ increase (Fig. 8h). CAPE above $2000\,\mathrm{J\,kg}^{-1}$ is restricted to the north of the MCS, UT relative humidity decreases to the northwest of the MCS (Fig. 8f). Convective activity at 9 km height is active at the location of the convective core. At 21:00 UTC, the convective core of the MCS is over Cape Verde, surrounded by STWL to the west,



south and north (Fig. 8j). The extent of cold pools has diminished, and is confined to the vicinity of the convective cells. The
low-level cyclonic circulation aligns with the convective core at 4 km height. Patches of CIN exceeding 1 J kg$^{-1}$ are located to
the west and north of the convective core, as well as further northeast and east (Fig. 8k). CAPE above 2000 J kg$^{-1}$ is restricted
to the northwest of the MCS. In this sector, AEJ speed is at its highest, resulting in a wind shear above 20 m s$^{-1}$ between 4 and
10 km height (not shown). This strong wind shear likely explains the almost complete absence of convective cells. The area
of DCCs diminishes and the overlap of MCS with UT dry air to the northwest (Fig. 8l). Convective cells up to 9 km in height
persists only in the convective core.

    Distinct convective regimes are simulated, transitioning from organized intense convection with linear shaped low-level
convergence at 06:00 UTC to more scattered cells reducing into a primary cluster at the MCS leading edge during the second
part of the day. Saharan air intrusions at STWL levels into the convective area of PH appears at all four times and increases
throughout the day, mostly to the west and southwest of the cloud, suggesting that the weak southwesterly wind circulation
identified Fig. 7b is decisive for this process. The intrusion of Saharan air into the convective area caused a separation between
the active cells to the north and those to the south, as evidenced by a tongue of low CAPE and high CIN values. This resulted
in a reduction of the southern convective activity from 11:00 UTC and to the dissipation of southern DCCs by 21:00 UTC.
Convective cold pools are present southeast of the convective area until 16:00 UTC. Saharan air and cold pools both coincide
with CIN values above 1 J kg$^{-1}$, suggesting that they both contribute to the convective inhibition.

**4.2   Contribution of STWL and cold pools to convective inhibition as seen in skew-T diagrams**

The contribution of STWL and cold pools to convective inhibition at 12:00 UTC is further assessed with skew-T diagrams
computed from the observed (Fig. 9a, d) and simulated (Fig. 9b, e) D4 and D5 profiles, and from simulated STWL and cold
pool atmospheric profiles averaged over D150 (Fig. 9c, f). Dropsonde D4 probes the atmosphere southwest of the MCS, a
location where STWL is simulated (Fig. 8d). The virtual temperature of 300 K at sea level gradually decreases until 870 hPa,
where a relatively warm and dry layer is observed, as indicated by a decrease in dew point temperature and increase in virtual
temperature at this level (Fig. 9a). At 330 hPa, the dew point temperature reduces by 10 K, indicating the presence of an UT dry
air layer. The warm and relatively dry air near the surface raises the lifting condensation level (LCL) to 970 hPa, which creates
favorable conditions for large CAPE (1237 J kg$^{-1}$) and weak CIN (0 J kg$^{-1}$). The profile is reproduced in the simulation, but
with a slightly lower surface virtual temperature, which reduces the CAPE to 1070 J kg$^{-1}$ and increases the CIN to 3.9 J kg$^{-1}$.
Dropsonde D5 probes the atmosphere inside the MCS (Fig. 9d). The measured virtual temperature and dew point temperature
at the surface are equal to 296.5 K, suggesting the presence of a cold pool. At 820 hPa, the dew point temperature decreases by
7 K and temperature increases by 3 K leading to CIN above 10 J kg$^{-1}$ and CAPE around 350 J kg$^{-1}$. The low surface virtual
temperature creates favorable conditions for large CIN and weak CAPE, but the low altitude of the LCL induced by the surface
moist environment partly balance this effect. The simulation reproduces the low surface virtual temperature and the low LCL,
345   although the air at 820 hPa is not as dry and warm as observed (Fig. 9e). In any case, the simulated CAPE and CIN values are
similar to those of D5.





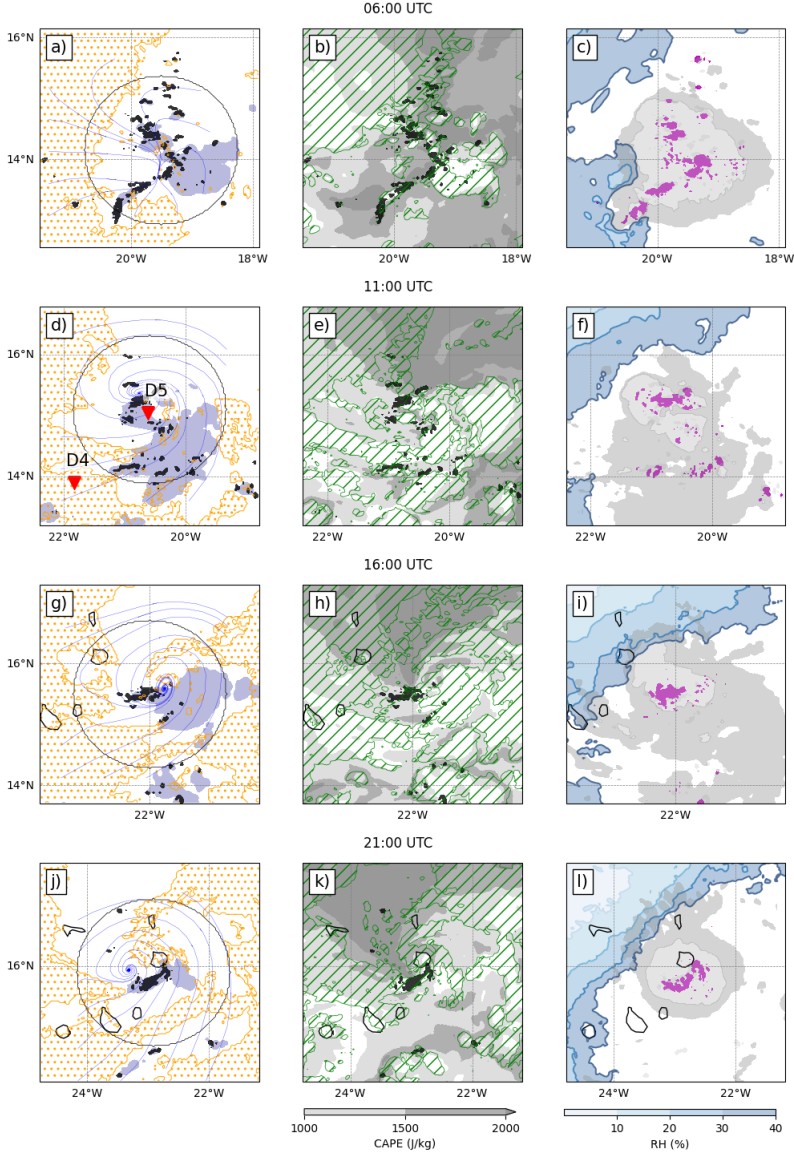

**Figure 8.** Horizontal cross sections at **(a, b, c)** 06:00, **(d, e, f)** 11:00, **(g, h, i)** 16:00 and **(j, k, l)** 21:00 UTC in the simulation. The black shading shows the vertical wind speed at 4 km height beyond 1 m s$^{-1}$ and the thin black lines the Cape Verde coastlines. Left panels: STWL (dotted orange area when relative humidity is below 80% and dust concentration is beyond 10 cm$^{-3}$, between 0.8 and 2 km height), cold pools (dark blue when buoyancy is below -0.03 m$^2$s$^{-1}$ at 50 m height), streamlines at 500 m height (blue lines) and radius of 150 km visually centered on the main convective cell at 4 km height (black circle). Middle panels: CAPE (gray shading) and CIN (green hatches when beyond 1 J kg$^{-1}$). Right panels: 10.8 $\mu$m brightness temperature between 210 and 230 K (gray) and below 210 K (light gray), relative humidity averaged between 8 and 11 km height below 40% (blue shading) and vertical wind speed at 9 km height beyond 1 m s$^{-1}$ (purple shading). In **(d)**, the red triangles show the D4 and D5 locations.





Skew-T diagrams from simulated STWL and cold pool atmospheric profiles show their respective impact on CAPE and CIN (Fig. 9c,f). At 12:00 UTC, the D150 area with CIN greater than $1\,\mathrm{J\,kg^{-1}}$ is covered by 20% STWL and 18% cold pools. Over these regions, the mean CAPE values are $914\,\mathrm{J\,kg^{-1}}$ for STWL and $577\,\mathrm{J\,kg^{-1}}$ for cold pools, while the mean CIN values are 350 $8.8\,\mathrm{J\,kg^{-1}}$ and $9.1\,\mathrm{J\,kg^{-1}}$, respectively. Most of the STWL is located to the west and south of D150, while the cold pools are mainly within the D150 area. The CAPE for the cold pools is relatively low compared to the CAPE for the STWL due to the lower virtual temperature of the moist adiabat used to calculate the CAPE.

Our results provide observational evidence of the distinct mechanisms between STWL and cold pools leading to CIN above $1\,\mathrm{J\,kg^{-1}}$. On the one hand, the CIN caused by STWL is a consequence of (i) the rise in altitude of the LCL due to dryer air in 355 the STWL and below, and (ii) the increased virtual temperature at low tropospheric levels. This finding is consistent with Wong and Dessler (2005) outcomes based on reanalysis data. On the other hand, the CIN caused by cold pools is a consequence of the low temperature and high humidity above the surface, which lower the LCL. Similarities between observations and the simulation give high confidence in our analysis of the impact of STWL and cold pools on convection.

### 4.3 UT dry air impact on the MCS PH vertical structure

360 A layer of dry and stable air layer is observed between 7 and $10\,\mathrm{km}$ height above the SAL along F7 (Fig. 3a, c, f, h) and in the anvil of the MCS along F8 (Fig. 10a) by the Falcon 20 relative humidity probe and dropsondes. Reflectivity and relative humidity sampled along $500\,\mathrm{km}$ of F8 are shown and compared to a similar $500\,\mathrm{km}$ long section in the simulation (Fig. 10a, b). The simulated vertical section is derived along a trajectory that closely aligns that of the aircraft, with adjustments to account for the positional discrepancies between the observed MCS and the simulated one. Reflectivity values above -15 dBZ are 365 observed up to $7\,\mathrm{km}$ height during the first $340\,\mathrm{km}$ and up to $14\,\mathrm{km}$ height in the MCS from 350 to $500\,\mathrm{km}$. Relative humidity values at $10\,\mathrm{km}$ height are below 10 % during the first $155\,\mathrm{km}$ and below 20% up to $325\,\mathrm{km}$. Relative humidity measured by dropsonde D6 at $240\,\mathrm{km}$ shows a steep vertical gradient of 50% around $8.5\,\mathrm{km}$ height. Observed reflectivity values are below -15 dBZ between 350 and $385$ km height from 6 to $11\,\mathrm{km}$ height (Fig. 10a). This feature is reproduced in the simulation with no reflectivity above -15 dBZ from 300 to $385\,\mathrm{km}$, between 7.5 and $11\,\mathrm{km}$. These low reflectivity values are at the same 370 altitude as the UT dry air, in both observation and simulation, suggesting an impact of this air on the MCS vertical extent. The observed tongue of low reflectivity values in the mid-troposphere is longer in F8 ($35\,\mathrm{km}$) than in F7 ($15\,\mathrm{km}$), in accordance with the increase area of UT dry air with time in D150, as shown at 16:00 and 21:00 UTC (Fig. 8i, l). Note that simulated dust concentrations above 2 and $5\,\mathrm{cm^{-3}}$ reach altitudes of respectively 8 and $14\,\mathrm{km}$ at $480\,\mathrm{km}$, at the location of a convective cell (Fig. 10b) showing that updrafts act as pumps lifting dust from the lower levels to the upper troposphere.

### 375 4.4 Summary

The evolution of MCS PH from 01:00 UTC September 11 to 00:00 UTC September 12 can be divided into three phases based on the coverage of HCCs, DCCs, CIN above $1\,\mathrm{J\,kg^{-1}}$ and UT dry air (Fig. 11). The areas covered by HCCs and DCCs are calculated over D300, a disk with a $300\,\mathrm{km}$ radius centered on the DCCs to take account of the large cloud extent of PH (Fig. 11a). The areas covered by CIN above $1\,\mathrm{J\,kg^{-1}}$ and UT dry air (with relative humidity below 40% averaged between 8



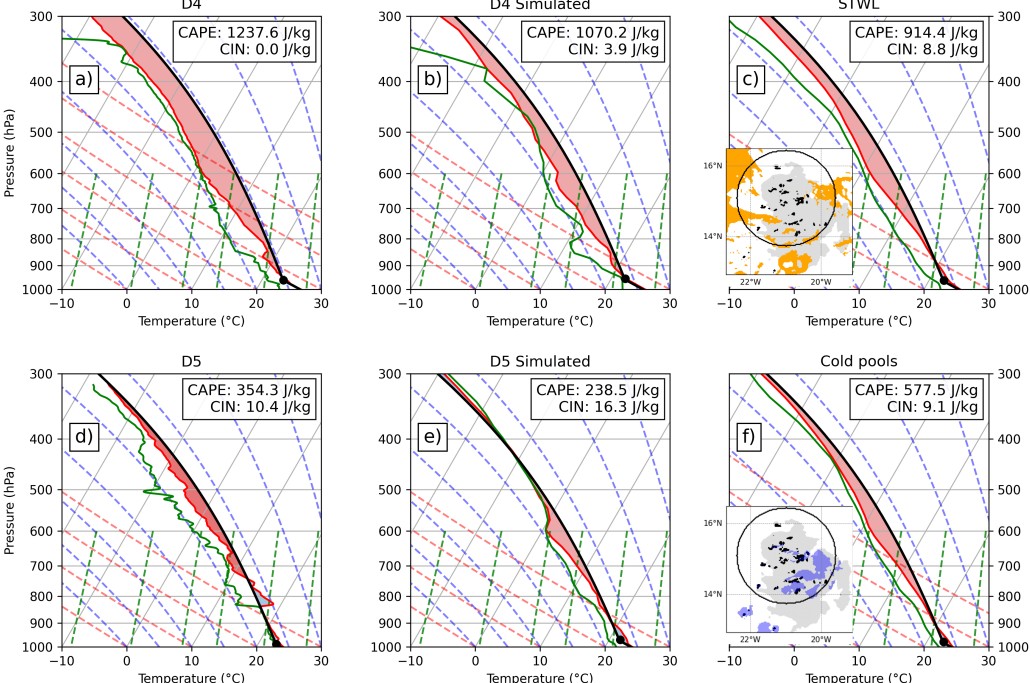

**Figure 9.** Skew-T diagrams of **(a)** dropsonde D4 launched at 11:45 UTC, **(b)** the profile simulated at 12:00 UTC at D4 location, **(c)** the average profile simulated at 12:00 UTC at STWL locations, **(d)** dropsonde D5 launched at 12:10 UTC, **(e)** the profile simulated at 12:00 UTC at D5 location, and **(f)** the average profile simulated at 12:00 UTC at cold pool locations. The black, red and green curves show the theoretical and sampled virtual air temperature and the dew point temperature, respectively. The red shading shows the CAPE and the black dot the lifting condensation level. Insets of **(c)** and **(f)** show the STWL (orange) and cold pools (blue) locations, respectively, from which average profiles are calculated, along with the vertical wind speeds at 5 km height beyond 1 m s$^{-1}$ (black shading) and reflectivity beyond -15 dBZ at 9 km height (gray shading) and a 150 km radius circle centered on the core of the MCS.

and 11 km height) are calculated over D150, close to the core of PH. Areas of CIN above 1 J kg$^{-1}$ covered by STWL and cold pools are attributed to the two terms, respectively. Note that only 1.6% of CIN above 1 J kg$^{-1}$ is covered by both STWL and cold pools.

The intense phase occurs from 00:00 to 07:00 UTC. The MCS is the most intense as shown by the fluctuation of HCCs up to 7% in the observation and almost 5% in the simulation. This results in a strong increase in the areas covered by DCCs and by CIN above 1 J kg$^{-1}$ due to cold pools. At the same time, the coverage of CIN above 1 J kg$^{-1}$ due to STWL and that of UT dry air gradually increase from relative low values. The mature phase takes place between 07:00 and 15:00 UTC. Convective activity is reduced as shown by the decline of HCCs, while DCCs continue to increase. This indicates a lag between the convective peak and the extent of the cloud shield. CIN above 1 J kg$^{-1}$ increases by up to 60%. This is due firstly to the expansion of cold pools, with a peak reaching half the CIN coverage at around 09:00 UTC. Then, areas of CIN above 1 J kg$^{-1}$



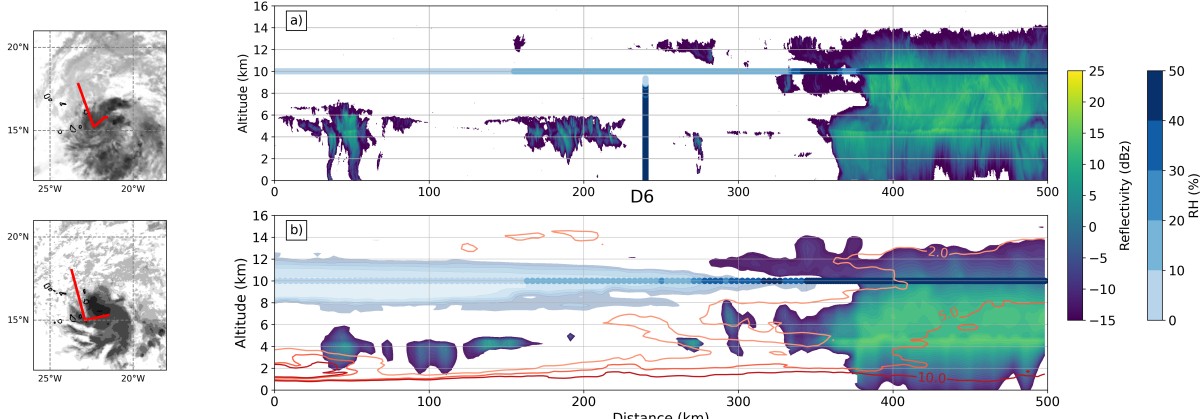

**Figure 10.** Right panels: **(a)** Vertical cross-section of RASTA radar reflectivity values beyond -15 dBZ (color shading) and relative humidity of the Falcon 20 probe (blue shading) along 500 km of the afternoon flight F8. Relative humidity from dropsonde D6 appears in blue shading at 240 km. **(b)** Vertical cross section of simulated reflectivity values beyond -15 dBZ (color shading), relative humidity at 10 km height (blue shaded line), relative humidity (blue shaded when below 50 %), dust concentrations (red contours at 2, 5 and 10 $cm^{-3}$). Observations are taken between 16:06 and 16:42 UTC and simulation at 16:00 UTC. Left panels: observed (top) and simulated (bottom) brightness temperature at 10.8 $\mu$m at 16:00 UTC (gray shading) and locations of the cross sections shown in the right panel (red lines).

due to STWL increase significantly until contributing up to 40% of the total coverage at 14:00 UTC. UT dry air coverage varies with time, up to around 10% between 13:00 and 16:00 UTC. The decaying phase runs from 15:00 to 00:00 UTC. It is marked by a decrease in HCC activity followed by a peak in coverage of CIN above 1 J kg$^{-1}$ at 18:00 UTC, with 39% attributed to STWL and 20% to cold pools. The DCC coverage drops from 20% to less than 7%, showing dissipation of the MCS while the UT dry air coverage becomes significant, reaching 18%.

## 5   Conclusion

The contribution of the Saharan air layer (SAL) and other airmasses and processes in the failed cyclogenesis of MCS PH over Cap Verde is addressed using data from the CADDIWA campaign and a convection-permitting simulation made with the Meso-NH mesoscale model. The joint analysis of SAL characteristics in observations and the simulation aims to improve understanding of SAL dust effects on the MCS. On 11 September 2021, MCS PH formed over the Atlantic at around 00:00 UTC, underwent an intense phase until 07:00 UTC, took a northwest track toward the SAL during its mature phase, declined from 15:00 UTC and finally dissipated on 12 September. The simulation of MCS PH and its environment is shown to be realistic when compared with observations from airborne measurements from the two CADDIWA flights on 11 September, and with satellite imagery.

The cloud shield and updrafts of MCS PH, the SAL and the three factors leading to PH inhibition documented in this study are illustrated in a 3D schematic (Fig. 12). It summarizes the key inhibiting roles of the Saharan trade wind layer (STWL), cold





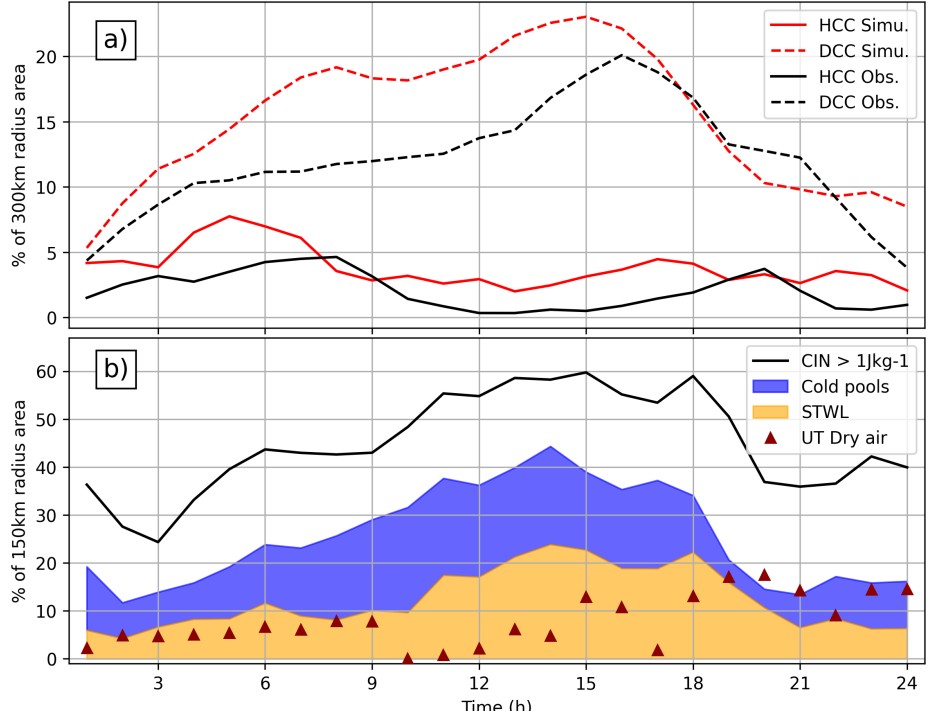

**Figure 11.** Time evolution between 00:00 UTC 11 September and 00:00 UTC 12 September of the surface area percentage of **(a)** simulated (red) and observed (black) cloud brightness temperature between 210 and 230 K (dashed lines) and below 210 K (solid lines) within the D300 disk and **(b)** CIN above 1 J kg$^{-1}$ (black line), CIN above 1 J kg$^{-1}$ attributed to STWL (orange shading), CIN above 1 J kg$^{-1}$ attributed to cold pools (blue shading) and relative humidity averaged between 8 and 11 km height below 40% (dark red triangles) within the D150 disk.

pools and upper-tropospheric (UT) dry air. It is worth noting that MCS PH is preserved from the intrusion of SAL air. Located between 2 and 4.8 km altitude, the SAL is advected away from PH, westward, by the African easterly wave (AEW) in which PH is embedded. The AEW circulation acts like a marsupial pouch, protecting PH from SAL intrusion.

The STWL is a warm, dry layer between 0.8 and 2 km containing aerosols with size distributions comparable to that of the
SAL. It is observed using lidar, dropsondes and microphysical probes, at trade wind levels, which justifies the name STWL. It differs from the transition layer between the MBL and the SAL, often referred to as a moist, clean trade wind inversion layer. The STWL is sampled at the southern edges of the SAL and beyond. Karyampudi et al. (1999) documented a connection between the SAL and the MBL south of a dust plume near Cap Verde (their Fig. 22). They hypothesized that this residual aerosol layer could result from dry deposition, turbulent mixing induced by vertical wind shear, and upward transport of
sea salt aerosol by cumulus clouds. The high resolution of CADDIWA observations provides new insights into the vertical structure of SAL boundaries, challenging the conceptual model of summer SAL and raising questions regarding dust transport and mixing over the Atlantic. In particular, the occurrence rate of STWL events is a topic that warrants further exploration.



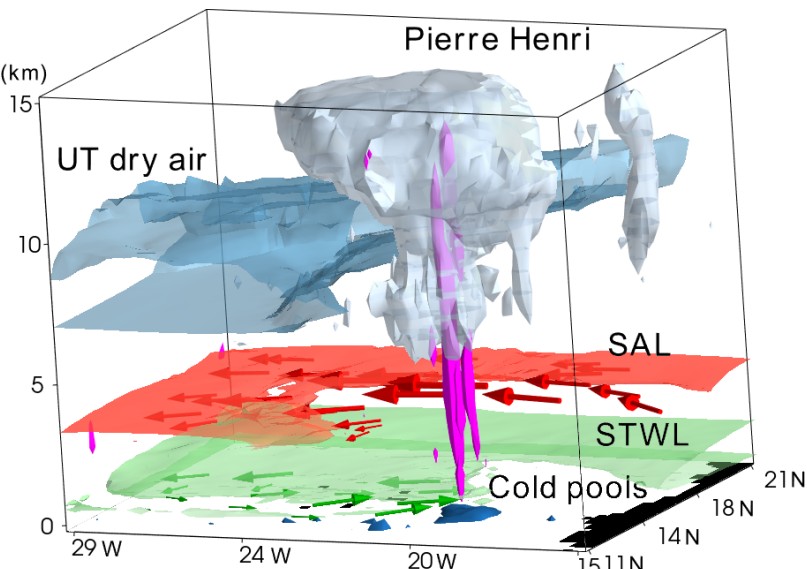

**Figure 12.** Three-dimensional schematic of MCS PH and its environment illustrated with simulation outputs at 11:00 UTC. The icy cloud shield of MCS PH appears in grey, updrafts in magenta, cold pools in dark blue, UT dry air in light blue, and the topography in black. For the sake of clarity, STWL and SAL dust contents are shown at a single level, here at 1 and 3.5 km, by green and red shading, respectively, while their wind speeds are indicated with green and red arrows.

The STWL is the primary inhibiting factor for MCS PH. The weak low-level convergent circulation facilitates its intrusion from the southwest into the PH convective core. The combination of dry low-level air elevating the lifting condensation level and warm air at STWL levels favors CIN by raising the level of free convection. The progressive intrusion of the STWL into PH during its intense and mature phases results in an increase in the areas of large CIN values within a radius of 150 km centered on the PH convective core. The 40% peak in CIN areas attributed to the STWL coincides with the overall decrease in PH convective activity, and more specifically with its dissipation in its southern part.

Cold pools are a second convective inhibition factor observed during CADDIWA. Thanks to downdrafts bringing cold air into the boundary layer, cold pools enhance atmospheric stability and raise the level of free convection, therefore strengthening the CIN and reducing the CAPE. The area of cold pools is maximal after the intense phase of PH, with almost half of the simulated CIN attributed to cold pools within a radius of 150 km centered on the PH convective core. The surface extent and prolonged duration of the cold pools may be due to low surface heat fluxes, which maintain the negative temperature perturbation. The identification of the STWL and cold pools allow us to attribute around two-third of the CIN to these two features. The portion of CIN that remains unattributed may, in part, result from limitations in the detection methods used for identifying the STWL and cold pools. A detailed characterization of the STWL and cold pools by future campaigns would help to better assess their role in the fate of the MCSs.



UT dry air is the third inhibiting factor for MCS PH. It is identified by a relative humidity of less than 15% between 7 and 11 km altitude along the northwestern edge of PH. The intrusion of dry air into the MCS progressively hinders the expansion

of its cloud anvil and the vertical development of its updrafts. CADDIWA observations from the afternoon flight show that the intrusion of dry air extends as long as 35 km into the MSC. At the mesoscale, UT dry air coverage increases within a 150 km radius from PH convective core over the course of the day, reaching 18% during the dissipation phase. It is associated with low vertical wind shear, suggesting the absence of an upper-level trough, contrary to previous studies (e.g., Arnault and Roux, 2011). These results underline the need for further research into the impact of UT dry air on cyclogenesis. Perspectives

include microphysical processes, such as the sublimation of the ice cloud anvil, mixing processes within the cloud anvil, or the inhibition of convective activity caused by the entrainment of dry air.

The realistic Meso-NH simulation of three convective inhibition processes, supported by CADDIWA observation, provides a new framework for the analysis of cyclogenesis in the Cape Verde region. Although the size distribution of dust aerosols is simplified and their radiative impacts omitted, the simulation effectively captures the key inhibition processes that led to the

dissipation of MCS PH. Future research using the Meso-NH model will take advantage of its LIMA two-moment microphysical scheme to analyze the sensitivity of convective processes and cloud properties to mineral dust parametrizations.

*Data availability.* The dropsondes data are available via DOI https://doi.org/10.25326/375 (Flamant, 2022b), the F20 thermodynamic and dynamic core data via DOI https://doi.org/10.25326/374 (Flamant, 2022a), the 2D-S probe via DOI https://doi.org/10.25326/651 (Coutris et al., 2022a), the UHSAS probe data via DOI https://doi.org/10.25326/652 (Coutris et al., 2022c), and the FSSP probe data via DOI

https://doi.org/10.25326/653 (Coutris et al., 2022b). The RALI data were downloaded from the https://rali.aeris-data.fr/ RALI project, 2023, specifically the RALI TC from https://rali.aeris-data.fr/catalogue/?uuid=1f4cb552-17f4-45e9-8ef7-fa980f7ff0f2 and the RASTA wind from https://rali.aeris-data.fr/catalogue/?uuid=39b9e681-1658-4083-8b74-27b6292fbbb8), the MODIS data from the Giovanni web portal (http://disc.sci.gsfc.nasa.gov/giovanni/, NASA, 2023) and the MSG data from ICARE (https://www.icare.univ-lille.fr/, AERIS/ICARE Data and Services Center, 2023).

*Author contributions.* GF performed the simulation and the analyzes under the supervision of JPC and TD. JD provided the RALI observations, PC the microphysical observations and all authors prepared the paper.

*Competing interests.* The authors declare that they have no conflict of interest.

*Acknowledgements.* This work was performed using HPC resources from GENCI at TGCC (grant 2023-A0140111437). CADDIWA data were collected using instruments from the French Airborne Measurement Platform, a national instrumental facility partially funded by
CNRS/INSU and CNES and operated by OPGC.





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
