# Peer review of "Failed cyclogenesis of a mesoscale convective system near Cape Verde: The role of the Saharan trade wind layer among other inhibiting factors observed during the CADDIWA field campaign"

_EGUsphere, 2025_

## Author Response (AR1)

We thank the Referee for his/her time and his/her constructive comments. We have complied with most of the proposed changes. In the following, the comments made by the Referee appear in black, our replies are in blue and additional text included in the revised MS appear in red.

This article presents a study on the cyclogenesis as observed and modelled after a Caddiwa field campaign case study. The goal is to explain why a Mesoscale Convective System (MCS) failed to produce a cyclogenesis often observed over the Atlantic sea under the wind of the Saharan air layer (SAL). Using aircraft measurements data and convection-permitting simulation run with the Meso-NH model, the authors show that some African easterly waves inhibit the impact of the SAL on the MCS. The article concludes that observations confirm the factors already known. The added value of the study is that it quantifies these factors in a specific case.

The article is clear and well written, even if it is full of acronyms and numbers that make it hard to understand the sentence and what the authors are trying to show. It can be accepted for publication after answering a few questions.

Major comments:

The main comment is the status of a single 'studied case' of this work. It is interesting to analyse the observations and the model in detail in order to quantify the impact of the various processes involved in maintaining cyclogenesis or not. But is this a specific case or a more general conclusion? To what extent can this study and its conclusions be considered generalizable? And if not, what more systematic data than a measurement campaign would enable this categorisation to be made more systematically? Is the present analysis confirmed by other cases where the cyclogenesis not failed and for the reasons proposed in this study? (other already published articles about cyclogenesis in this region?) At the end of the conclusion, we added: "This study examines the failed cyclogenesis of a single MCS. As such, it remains uncertain how representative our findings are for other storms forming in the Cape Verde region. In this case, warm and dry Saharan air intruded to the southwest of the MCS — an area previously identified as unfavorable for cyclone intensification in the presence of SAL dry air (Shu and Wu, 2009). However, in that study, SAL dry air was characterized using relative humidity between 600–700 hPa, and not within the STWL. It would therefore be valuable to refine this type of systematic analysis by focusing specifically on the STWL. Moreover, Shu and Wu (2009) investigated the weakening of named tropical cyclones. In contrast, there is as yet no comprehensive study of failed cyclogenesis events, such as the one presented here for a single case. Such a study is needed to better understand the wider implications of our results, particularly with regard to the influence of STWL. A crucial aspect of such a study would be to document the vertical distribution of dust at high resolution. Rather than relying solely on lidar data obtained during a field campaign, systematic monitoring of the STWL could be carried out using ground-based or space-based lidar systems. Combined with back-trajectory analysis, aerosol observations could be potentially identified as dust, depending on aerosol origin. In addition, it is essential to monitor the dryness of the STWL with specific measurements, as this is an essential characteristic of this layer."

Finally, in a forecasting context, is it possible to set thresholds for the various factors and thus know in advance whether cyclogenesis will take place or not? Forecasting is beyond the scope of our paper.

Minor comments:

l.12 Acronyms may be useful but too much acronyms make reading difficult. Only in the abstract: MCS, CADDIWA, Meso-NH, PH, SAL, STWL, CIN, UT. For ease of reading, we chose to write Pierre Henri instead of PH and upper-tropospheric dry air instead of UT. The acronyms MCS, SAL and CIN (as well as AEW, AEJ, CAPE) are relatively well known in the community, so we decided to keep them as they are. Meso-NH is designed as the model name. We decided to add "field campaign" after the acronym CADDIWA in the text in case the reader skipped the introduction. The acronym STWL is written 26 times, we chose to keep it to avoid weighing down the text with too many repetitions.

l.25: "... on going debate (Shu and Wu, 2009)". Very long debate. A more recent reference? The references to the other studies cited in the paragraph, that is Luo and Han (2021), Pan et al. (2018), Sun and Zhao (2020), Xian et al. (2020), have been added.

l.4 A "protective shell" in place of 'marsupial pouch'?? is this a common way of describing this atmospheric phenomenon? The marsupial pouch, also known as the "pouch theory", is a common theory, but in a specific community. Proposed by Dunkerton et al. (2009), it refers to a region of cyclonic recirculation relatively protected from lateral intrusion in the parent wave of a MCS. To clarify the role of this protective pouch in the abstract, we changed the sentence "forms a marsupial pouch" with "forms a region of cyclonic recirculation relatively protected from lateral intrusions called marsupial pouch".

l.61: the 'respective' role? because the fact they have a role seems to be already known. Added.

l.82 It is necessary to have a top domain at 26 km ASL for this kind of study? Would not it be more interesting, for the same calculation cost, to have a lower top and therefore a thinner first vertical cell depth than 30m? Is 24h enough for the spin-up (i.e absorb the impact of the large scale due to the initial conditions)?
We added "The vertical grid has [...] and a sponge layer in the last 8 km to damp gravity waves." This explains why the top of the model is much higher than the top of the deep convective clouds. At the base of the model, the first vertical cell depth of 30 m is sufficient to represent the marine boundary layer.
In Fig. 11, we used hourly Meso-NH outputs starting at 01:00 UTC. So we implicitly assumed a spin-up time of less than 1 hour. The comparison between simulated and observed cloud brightness temperature between 210 and 230 K and below 210 K within the D300 disk is realistic enough. This shows that 1 hour is a sufficient time for the model to develop its own cloud fields.

l.205 Again about initial conditions: why not use the same model as input knowing forecast was made for the previous days? The domain, start and duration of the simulation for this case study and the forecast made during the CADDIWA campaign differ. We take advantage of the availability of the back-trajectories calculated from the forecasts to use them to identify the origin of the STWL and SAL.

Figure 5: what is the unit for the dust concentrations in cm-3? number of particles/cm3? mass per cm3? a concentration is 'something' per 'volume'. Thank you for noticing this misleading unit. We changed it to #/cm3, and in Fig. 6 as well.

l.242 what is the meaning of an 'accumulation mode' for dust? Is it possible than other aerosols are present in the studied plume? The meaning of "accumulation mode" refers to the range of aerosol size distribution (0.1 to 1 $\mu$m), added in the revised version). This is a standard definition of aerosol size distribution, which also applies to dust. For example, "aerosols less than 1 $\mu$m are considered fine aerosols, and are divided into Aitken mode ($<0.1\,\mu$m ) and accumulation mode ($>0.1\,\mu$m). Particles larger than about 1 $\mu$m are often referred to as coarse aerosols" (Mahowald et al., 2014). Note that, "assuming that the SAL contains mainly dust aerosols" (l.242) means that it is possible than other aerosols are present in the studied plume.

l.244 What is the definition of fine and coarse aerosol? To clarify, the sentence is now "These three dust layers differ, however, in their size distribution for the fine ($<1\,\mu$m) and coarse ($>1\,\mu$m) aerosol modes."

l.288 what is the reference for this treshold of 10 ?/cm-3 to define the STWL? To identify the STWL we tested several of its properties, alone or in combination, such as high static stability, low relative humidity and high dust concentration. We finally chose the 10 #/cm3 dust concentration threshold because of its simplicity and intuitive representation as a marker of Saharan air. To clarify, we added "The threshold of $10\,\text{cm}^{-3}$ was chosen visually to best align with changes in both stability and relative humidity in the simulation domain. Its relatively high value means that the probability of overestimating STWL coverage is low."

**References**

Dunkerton, T. J., Montgomery, M., and Wang, Z.: Tropical cyclogenesis in a tropical wave critical layer: Easterly waves, Atmospheric Chemistry and Physics, 9, 5587–5646, 2009.

Luo, H. and Han, Y.: Impacts of the Saharan air layer on the physical properties of the Atlantic tropical cyclone cloud systems: 2003–2019, Atmospheric Chemistry and Physics, 21, 15 171–15 184, 2021.

Mahowald, N., Albani, S., Kok, J. F., Engelstaeder, S., Scanza, R., Ward, D. S., and Flanner, M. G.: The size distribution of desert dust aerosols and its impact on the Earth system, Aeolian Research, 15, 53–71, 2014.

Pan, B., Wang, Y., Hu, J., Lin, Y., Hsieh, J.-S., Logan, T., Feng, X., Jiang, J. H., Yung, Y. L., and Zhang, R.: Impacts of Saharan dust on Atlantic regional climate and implications for tropical cyclones, J. Climate, 31, 7621–7644, https://doi.org/10.1175/JCLI-D-16-0776.1, 2018.

Shu, S. and Wu, L.: Analysis of the influence of Saharan air layer on tropical cyclone intensity using AIRS/Aqua data, Geophys. Res. Lett., 36, L09 809, https://doi.org/10.1029/2009GL037634, 2009.

Sun, Y. and Zhao, C.: Influence of Saharan dust on the large-scale meteorological environment for development of tropical cyclone over North Atlantic Ocean Basin, J. Geophys. Res. Atmos., 125, e2020JD033 454, https://doi.org/10.1029/2020JD033454, 2020.

Xian, P., Klotzbach, P. J., Dunion, J. P., Janiga, M. A., Reid, J. S., Colarco, P. R., and Kipling, Z.: Revisiting the relationship between Atlantic dust and tropical cyclone activity using aerosol optical depth reanalyses: 2003–2018, Atmos. Chem. Phys., 20, 15 357–15 378, https://doi.org/10.5194/acp-20-15357-2020, 2020.

We thank the Referee for his/her time and his/her constructive comments. We have complied with most of the proposed changes. In the following, the comments made by the Referee appear in black, our replies are in blue and additional text included in the revised MS appear in red.

This study investigates the inhibiting factors that led to the failed cyclogenesis of the mesoscale convective system (MCS) Pierre Henri (PH), observed during the CADDIWA field campaign. Using high-resolution airborne observations and a convection-permitting Meso-NH model, the authors analyze the interactions between the Saharan Air Layer (SAL), the Saharan Trade Wind Layer (STWL), cold pools, and upper tropospheric (UT) dry air. The study finds that: 1) the STWL, a relatively underexplored feature, played a significant role in increasing convective inhibition (CIN), contributing up to 40% of CIN during the mature phase of PH; 2) cold pools generated by convection further suppressed cyclone development, with their CIN contribution peaking at 50% post-intense phase; and 3) UT dry air limited the MCS's anvil expansion, with relative humidity below 15% between 7 and 11 km altitude, covering 18% of the MCS environment during dissipation. The findings are well-supported by observational data and validated against numerical simulations. I only have two minor comments for the authors to consider.

The study focuses on the failed cyclogenesis of one MCS (PH), but it is unclear how representative these findings are for other storms in the Cape Verde region. It would be interesting to more comprehensively compare the findings in this study with past ones on failed and/or successful Cape Verde cyclogenesis cases to improve the broader implications. At the end of the conclusion, we added: "This study examines the failed cyclogenesis of a single MCS. As such, it remains uncertain how representative our findings are for other storms forming in the Cape Verde region. In this case, warm and dry Saharan air intruded to the southwest of the MCS — an area previously identified as unfavorable for cyclone intensification in the presence of SAL dry air (Shu and Wu, 2009). However, in that study, SAL dry air was characterized using relative humidity between 600–700 hPa, and not within the STWL. It would therefore be valuable to refine this type of systematic analysis by focusing specifically on the STWL. Moreover, Shu and Wu (2009) investigated the weakening of named tropical cyclones. In contrast, there is as yet no comprehensive study of failed cyclogenesis events, such as the one presented here for a single case. Such a study is needed to better understand the wider implications of our results, particularly with regard to the influence of STWL. A crucial aspect of such a study would be to document the vertical distribution of dust at high resolution. Rather than relying solely on lidar data obtained during a field campaign, systematic monitoring of the STWL could be carried out using ground-based or space-based lidar systems. Combined with back-trajectory analysis, aerosol observations could be potentially identified as dust, depending on aerosol origin. In addition, it is essential to monitor the dryness of the STWL with specific measurements, as this is an essential characteristic of this layer."

While the study effectively shows that STWL increased CIN, the mechanism of STWL intrusion into the MCS and its modification of convective processes needs clearer explanation. I suggest further elaborating on how weak low-level circulation allows STWL intrusion based on the current analysis. We refer to the mechanism of STWL intrusion into the MCS when commenting Fig. 7b. To clarify the point, the sentences are now "At 1.5 km height, streamlines with speeds below $10\,\mathrm{m\,s^{-1}}$ and dust concentrations above $15\,\mathrm{cm^{-3}}$ are present from the west up to the southwestern convective cells (Fig. 7b). An almost closed circulation pattern with an approximate diameter of 40 km is at the center of the convective cells, which prevents the STWL intrusion. However, a weak converging circulation to the southwest of the MCS facilitates the intrusion of the STWL into the nearby convective cells, as shown by the presence of dust concentrations above $15\,\mathrm{cm^{-3}}$."
In the conclusion, the sentence lines 419-420 "The weak low-level convergent circulation facilitates its intrusion from the southwest into the convective core of MCS Pierre Henri." is now "Its intrusion from the southwest into the convective core of the MCS is facilitated by a weak low-level convergent circulation due to the absence of a marsupial pouch at this level."

**References**

Shu, S. and Wu, L.: Analysis of the influence of Saharan air layer on tropical cyclone intensity using AIRS/Aqua data, Geophys. Res. Lett., 36, L09 809, https://doi.org/10.1029/2009GL037634, 2009.